# Self-Supervised Predictive Coding with Multimodal Fusion for Patient Deterioration Prediction in Fine-grained Time Resolution

**Kwanhyung Lee**[1], **John Won**[1], **Heejung Hyun**[1], **Sangchul Hahn**[1], **Edward Choi**[2], **Joohyung Lee**[1*]

[1]AITRICS

[2]KAIST

{kwanlee9209,johnwon,alex,steve,chris}@aitrics.com,
edwardchoi@kaist.ac.kr

## Abstract

Accurate time prediction of patients' critical events is crucial in urgent scenarios where timely decision-making is important. Though many studies have proposed automatic prediction methods using Electronic Health Records (EHR), their coarse-grained time resolutions limit their practical usage in urgent environments such as the emergency department (ED) and intensive care unit (ICU). Therefore, in this study, we propose an hourly prediction method based on self-supervised predictive coding and multi-modal fusion for two critical tasks: mortality and vasopressor need prediction. Through extensive experiments, we prove significant performance gains from both multi-modal fusion and self-supervised predictive regularization, most notably in far-future prediction, which becomes especially important in practice. Our uni-modal/bi-modal/bi-modal self-supervision scored 0.846/0.877/0.897 (0.824/0.855/0.886) and 0.817/0.820/0.858 (0.807/0.81/0.855) with mortality (far-future mortality) and with vasopressor need (far-future vasopressor need) prediction data in AUROC, respectively.

## 1 Introduction

In the emergency department (ED) and intensive care unit (ICU), accurate time prediction of clinically critical events is crucial to make timely interventions for acutely deteriorating patients. Moreover, the early prediction of critical events enables precise prioritization and preparation for high-risk patients by an efficient resource allocation Wang & Lan (2022); Wu et al. (2017). As a result, many studies have reported their early prediction systems using Electronic Health Records (EHR) Wang & Lan (2022); Wu et al. (2017); Sung et al. (2021).

Reported studies, however, make predictions in coarse-grained time resolution: 1) predicting vasopressor need within 24/48 hour Wanyan et al. (2021); Choi et al. (2022) and within 6-10 hours Suresh et al. (2017) and 2) predicting mortality within 24/48 hours Wanyan et al. (2021) and 3) within the whole hospitalization period Wang & Lan (2022); Choi et al. (2019). However, prediction over a coarse-grained time resolution can be impractical where timely decision-making and rapid intervention are crucial. In this study, therefore, we aim to 1) make an hourly prediction over the future 12 hours and 2) enhance the far-future (early) prediction by adding static features to the EHR time-series data Wu et al. (2017).

Several studies suggest utilizing multi-modal EHR data in coarse-grained time resolution, such as Wang *et al.* Wang & Lan (2022), which predicts future mortality with physiological index, treatment records, and hospitalization records, or Suresh *et al.* Suresh et al. (2017), which predicts intervention needs with demographic data, vital signs/lab tests, and clinical notes. However, the benefit of additional modalities was uncertain, since these studies report the performance of the multi-modal model without comparing it to the performance of the individual (uni-modal) model.

---

[*]Correspondence to chris@aitrics.com

Our main contributions are as follows; 1) we propose a novel fine-grained time deterioration prediction method for two representative critical events in ED and ICU, i.e., mortality and vasopressor need; 2) we show our future encoding method with additional normalization is important in our self-supervised predictive regularization for fine-grained deterioration prediction (Fig.2); 3) through an extensive experiment, we show that both multi-modal fusion and our self-supervised predictive regularization improves the predictive performance, especially the far-future (early) prediction, which is crucial but more challenging than near-future prediction Wu et al. (2017); Danilatou et al. (2022).

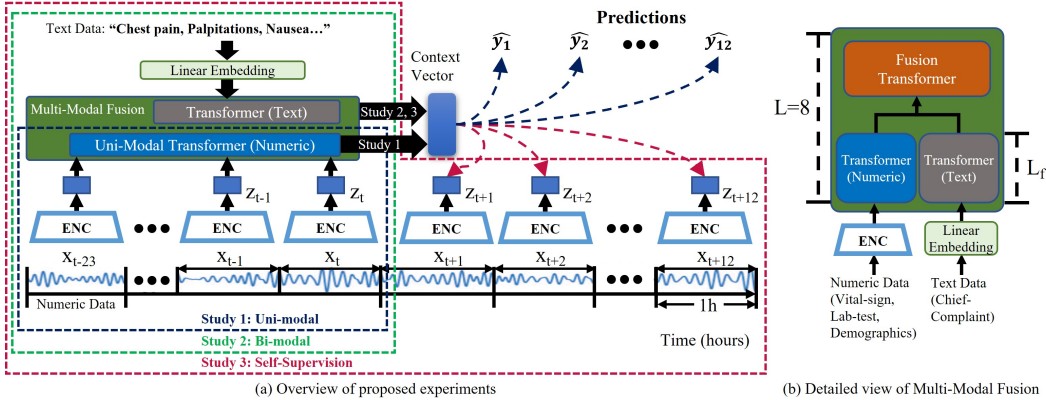

(a) Overview of proposed experiments          (b) Detailed view of Multi-Modal Fusion

Figure 1: (a) Overview of proposed 1) uni-modal, 2) bi-modal, 3) self-supervised methods. All methods include predictions (supervised learning). $t$ refers to the time when the prediction is made. (b) Fusion structure with $L_f$ indicating where the fusion starts.

## 2 METHODS

Fig. 1 illustrates the overall scheme of our network. In this paper, we propose a novel deterioration prediction model for a fine-grained time resolution. To this end, we gradually add features with the best performance in the following order; 1) we select a uni-modal model to learn EHR numeric data; 2) we compare bi-modal (numeric + text data) fusion strategies; 3) we compare various joint self-supervised learning (SSL) strategies; and 4) we compare various encoding methods for the future EHR numeric data to perform joint SSL. For each study, we select the best-performing one and fix it for the remaining studies to assess the efficacy of the added component, e.g., additional modality or SSL loss (Fig. 3). We conduct every study for both mortality and vasopressor need prediction data. At the end of the last study, therefore, we propose our deterioration prediction model for a fine-grained time resolution with the accumulated components. For a fair comparison, we fix all Transformer-based models to equally have 8 transformer layers, 4 multi-heads, and 256 feature dimensions (Fig. 1-(b)).

### 2.1 ELECTRONIC HEALTH RECORD DATA

To simulate an urgent hospital environment, we used the MIMIC-ED and MIMIC-IV (Medical Information Mart for Intensive Care in Emergency Department and IV) datasets Johnson et al.; 2020). Since both datasets share the same patients, we merged chief complaints from MIMIC-ED (text data) and a total of 18 different time-series features from MIMIC-IV (numeric data), i.e., vital signs, lab-test results, and demographic features (age and gender). Vital-sign includes heart rate, respiration rate, and 4 other items. Lab-test result include Hematocrit, Platelet, and 8 more items Sung et al. (2021). More detailed information about the selected features and their importance for our prediction tasks are summarized in Appendix section A.1 and A.2. We labeled the occurrence of mortality and vasopressor usage in binary. The sampling frequency for the time-series data is 1 hour, and we applied carry-forward imputation (most recent value of the past) for missing features. For vital-sign and lab-test features, we applied min-max normalization using the minimum and maximum values from the entire training dataset. The input time length varies from 3 to 24 hours to 1) challenge the prediction for patients shortly after admission and 2) simulate varying ED environments Henriksen

et al. (2014). We used zero-padding to fix the input data length as 24 hours and only considered patients who had ICU stays of 15 to 1440 hours.

Table 1: Data statistics with patient numbers for mortality prediction and vasopressor need prediction tasks.

| Tasks | Mortality | Vasopressor |
|---|---|---|
| Data Split | Train / Test | Train / Test |
| Positive Subjects | 2544 / 262 | 5827 / 606 |
| Negative Subjects | 24492 / 2836 | 21941 / 2580 |

To select the best-performing model for the four studies (Table 2), we used the averaged validation area under the receiver operating characteristic (AUROC) from 5-fold cross-validation (CV). To assess the efficacy of the additional component, i.e. text data and SSL, we compared the averaged test AUROC from the 5-fold CV of the best-performing uni-modal, bi-modal, and the model trained with SSL (Fig. 3).

## 2.2 Uni-modal Model for EHR Numeric Data

We explored four different models: GRU-D Che et al. (2018), LSTM, Transformer Vaswani et al. (2017), and Graph Transformer Choi et al. (2019); Lee et al. (2022). All four models map time-series numeric data (vital-signs, lab tests, demographics) $x_{\leq t} \in \mathbb{R}^{18 \times T_i}$, ($T_i = 24$ in our study) into a context vector $c_t \in \mathbb{R}^{256}$, which is then mapped to 12 probabilities for our 12-hour fine-grained time prediction. For mapping, we use 12 distinct 2-layer Multilayer perceptron (MLP) with batch normalization and ReLU non-linearities between the 2 linear layers, followed by a sigmoid function. Both GRU-D and LSTM receive the raw input $x_{\leq t}$, whereas both Transformer and Graph Transformer receive the encoded input $z_{\leq t}$, which is $x_{\leq t}$ encoded by the 2-layer MLP with Layer Normalization (LN) and ReLU activation, to output the context vector $c_t$ (CLS token vector) (Fig. 1-(a)).

## 2.3 Bi-modal Fusion Strategy for EHR Text Data

Alike the best-performing uni-modal model (Table 2), we use the vanilla Transformer with BERT tokenization Devlin et al. (2016) for EHR text data. We fuse the outcomes of the $L_f$-th layer of the text and numeric Transformers (Fig. 1-(b)); we refer the *early* and *mid* fusion to the fusions that occur after the 0-th (before Transformer) and 4-th layer of the Transformers of text and numeric data (Fig. 1-(b)). Note that we rigorously explore the *early* and *mid* fusion due to the poor performance of the *late* fusion (fusion after 9-th layer) during our preliminary experiment. Moreover, we experimented with three different types of fusion methods: 1) Multimodal Bottleneck Transformer (MBT) Nagrani et al. (2021), 2) Multimodal-Transformer (MT) Nagrani et al. (2021), 3) Bi-Cross Modal Attention Transformer (BCMAT) Tsai et al. (2019). MBT, MT, and BCMAT respectively utilize the fusion bottleneck (FSN) tokens Nagrani et al. (2021), concatenation, and attention fusion after the $L_f$-th layer. Specifically, MBT creates and lets two Transformers share four new FSN tokens after the $L_f$-th layer of the text and numeric Transformers. MT concatenates the outcomes of the $L_f$-th layer of both Transformers. BCMAT uses two parallel attention fusions; after $L_f$-th layer, one Transformer uses its outcome as both the key and value and the outcome of the other Transformer as the query, and vice versa for the other fusion.

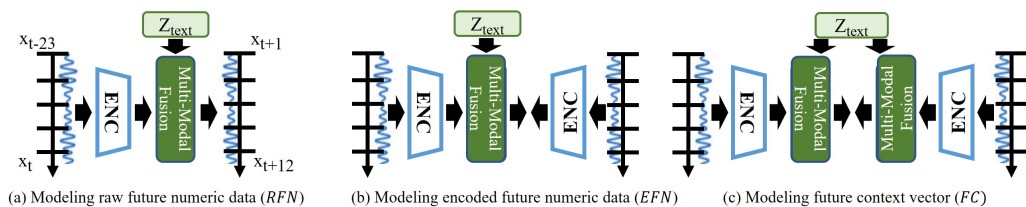

(a) Modeling raw future numeric data (*RFN*)  (b) Modeling encoded future numeric data (*EFN*)  (c) Modeling future context vector (*FC*)

Figure 2: Three different methods to encode the future numeric data ($x_{t+k}$) for self-supervision. (b) is the default method used in Section 2.4 and illustrated in Fig 1-a.

## 2.4 Self-supervised Regularization

For time-series data, Oord *et al.* Oord et al. (2018) introduced Contrastive Predictive Coding (CPC) which self-supervises networks to encourage capturing global information, i.e. 'slow feature', using Noise Contrastive Estimation (NCE). Moreover, Wanyan *et al.* Wanyan et al. (2021) and Zang *et al.* Zang & Wang (2021) proposed to regularize networks by adding a SSL loss to the supervision loss. Motivated by these studies, we regularize our bi-modal network using CPC whose loss can be described as Eq. 1:

$$\mathcal{L}_{NCE} = -log \frac{exp(z_{t+k}^T W_k c_t)}{\sum_j exp(z_j^T W_k c_t)} \tag{1}$$

$$\mathcal{L}_{cosine} = -\frac{z_{t+k}^T W_k c_t}{\|z_{t+k}\|_2 \cdot \|W_k c_t\|_2} \tag{2}$$

$$\mathcal{L}_{l_2} = \|z_{t+k} - W_k c_t\|_2 \tag{3}$$

However, since we add supervision loss to SSL loss (NCE), we assume the model will not converge to a trivial solution (model collapse) even if the SSL loss does not contain negative pairs. Therefore, we also implement cosine (Eq. 2) and $L_2$ loss (Eq. 3) alongside NCE. In the equations above, $c_t$ refers to the context vector (Fig. 1-(a)) from the bi-modal fusion Transformer. We use linear transformation $W_k c_t$ for SSL prediction where different $W_k$ is used for different time step $k$. In this study, we use 12 distinct $W_k$ to concurrently predict 12 encoded future numeric $z_{t+1}$, $z_{t+2}$, ... , $z_{t+12}$. Note that, the encoder for the past numeric $x_{\leq t}$ encodes 12 distinct future numeric $x_{>t}$ to $z_{>t}$ as well. Since we maximize the mutual information (MI) between linearly transformed context vectors and 12 distinct $z_{>t}$, which share the same encoder, the encoder is encouraged to learn the information shared across all time points; we assume this 'slow feature' encourages far-future prediction. For $L_2$ loss of $MT_{early}$ (mortality prediction), we explore additional LN to normalize $c_t$, because $L_2$ loss from unnormalized $c_t$ is large in its value compared to supervision loss ($MBT_{early}$ does not need additional LN since its context vector is already normalized).

## 2.5 Encoding Future Numeric Data for Self-supervision

The original CPC paper proposes to maximize MI between context vector $c_t$ and encoded future numeric $z_{t+k}$ (Fig. 2-(b)) instead of using raw future numeric $x_{t+k}$ (Fig. 2-(a)). The aim of the original CPC paper is to avoid modeling the high dimensional distribution of the raw data $x_{t+k}$. Since our raw data $x_{t+k}$ has lower dimensions than the encoded data $z_{t+k}$. Therefore, we hypothesized that modeling the raw future numeric may outperform modeling the encoded future numeric (Fig. 2-(a)). Lastly, we also experimented Fig. 2-(c) which encourages similarity between the context vectors of the past and the future assuming that using the time-aggregated context vector for SSL may outperform the others. We compare the performance of these three SSL structures (Fig. 2) in Sec. 3.2.

## 3 Results and Discussion

In this section, we analyze the performances of different 1) models to learn EHR numeric data, 2) strategies to fuse the encoded EHR text data and numeric data, 3) self-supervision loss, and 4) how to use the future EHR numeric data for self-supervision. We conduct these four studies to predict mortality and vasopressor need independently and illustrate the result in Table 2.

## 3.1 Transformer works better than other alternatives for learning EHR numeric data

As shown in Table 2, the vanilla Transformer outperforms all other alternatives to predict mortality and vasopressor need in a fine-grained time course. Note that the vanilla Transformer excels over the Graph Transformer suggesting that learning temporal relationships is more important than learning inter-feature relationships. All four models show gradual degradation in performance when predicting further in the future, which reflects the difficulty in far-future prediction.

Table 2: Validation AUROC of 1) uni-modal, 2) bi-modal, 3) self-supervision loss, and 4) different future numeric encoding methods for self-supervision. The range $a$~$a+1$ indicates future prediction's hourly range. Best performing option in average AUROC (bold) is selected and applied to the remaining studies to assess the efficacy of an additional feature. $*$ indicates additional normalization (Sec. 2.4). $RFN, EFN,$ and $FC$ refer to the different future numeric data encoding methods for SSL (Fig. 2).

| Models | 0~1 | 1~2 | 2~3 | 3~4 | 4~5 | 5~6 | 6~7 | 7~8 | 8~9 | 9~10 | 10~11 | 11~12 | Avg. |
|---|---|---|---|---|---|---|---|---|---|---|---|---|---|
| **Mortality Prediction (AUROC)** | | | | | | | | | | | | | |
| GRU-D | 0.897 | 0.867 | 0.852 | 0.838 | 0.831 | 0.822 | 0.812 | 0.798 | 0.788 | 0.801 | 0.786 | 0.78 | 0.823 |
| LSTM | 0.918 | 0.886 | 0.868 | 0.86 | 0.852 | 0.84 | 0.841 | 0.834 | 0.827 | 0.819 | 0.817 | 0.809 | 0.848 |
| **Transformer** | **0.921** | **0.892** | **0.877** | **0.865** | 0.856 | 0.846 | 0.84 | 0.837 | 0.827 | **0.823** | **0.823** | **0.821** | **0.852** |
| Graph Transformer | 0.903 | 0.885 | 0.875 | **0.865** | **0.858** | **0.847** | **0.842** | **0.839** | **0.832** | 0.822 | **0.823** | 0.819 | 0.851 |
| $MBT_{early}$ | 0.91 | 0.887 | 0.881 | 0.876 | 0.872 | **0.865** | 0.859 | 0.86 | **0.858** | 0.854 | 0.851 | **0.857** | 0.869 |
| $MBT_{mid}$ | 0.911 | 0.888 | 0.882 | 0.875 | **0.873** | 0.863 | 0.858 | **0.861** | 0.854 | **0.858** | **0.857** | **0.857** | 0.87 |
| $\mathbf{MT_{early}}$ | 0.913 | 0.897 | 0.888 | **0.88** | **0.873** | **0.865** | **0.863** | 0.86 | 0.854 | 0.856 | 0.852 | 0.848 | **0.871** |
| $MT_{mid}$ | **0.918** | 0.896 | 0.885 | 0.876 | 0.871 | 0.862 | 0.858 | 0.854 | 0.846 | 0.846 | 0.843 | 0.839 | 0.866 |
| $BCMAT_{early}$ | 0.916 | **0.9** | **0.89** | 0.873 | 0.868 | 0.863 | 0.853 | 0.855 | 0.844 | 0.84 | 0.84 | 0.838 | 0.865 |
| $BCMAT_{mid}$ | 0.903 | 0.888 | 0.876 | 0.869 | 0.864 | 0.854 | 0.847 | 0.848 | 0.841 | 0.842 | 0.845 | 0.842 | 0.86 |
| $MT_{early}NCE_{EFN}$ | 0.9 | 0.885 | 0.875 | 0.867 | 0.868 | 0.853 | 0.848 | 0.851 | 0.844 | 0.837 | 0.836 | 0.822 | 0.857 |
| $MT_{early}Cosine_{EFN}$ | 0.91 | 0.902 | 0.89 | 0.872 | 0.884 | 0.876 | 0.84 | 0.86 | 0.865 | 0.875 | 0.855 | 0.839 | 0.872 |
| $MT_{early}L2_{EFN}$ | 0.696 | 0.562 | 0.75 | 0.709 | 0.687 | 0.722 | 0.736 | 0.574 | 0.647 | 0.667 | 0.676 | 0.696 | 0.677 |
| $\mathbf{MT_{early*}L2_{EFN}}$ | **0.926** | **0.907** | **0.898** | **0.885** | **0.89** | **0.883** | **0.871** | **0.879** | **0.877** | **0.876** | **0.88** | **0.871** | **0.887** |
| $\mathbf{MT_{early*}L2_{RFN}}$ | **0.902** | **0.904** | **0.895** | **0.887** | **0.895** | **0.892** | **0.886** | **0.885** | **0.883** | **0.875** | **0.891** | **0.878** | **0.889** |
| $MT_{early*}L2_{FC}$ | 0.921 | 0.894 | 0.885 | 0.874 | 0.868 | 0.858 | 0.854 | 0.849 | 0.847 | 0.841 | 0.836 | 0.829 | 0.863 |
| **Vasopressor Need Prediction (AUROC)** | | | | | | | | | | | | | |
| GRU-D | 0.819 | 0.815 | 0.813 | **0.813** | **0.811** | 0.805 | 0.801 | 0.8 | 0.797 | 0.794 | **0.795** | 0.789 | 0.804 |
| LSTM | 0.814 | 0.81 | 0.808 | 0.806 | 0.802 | 0.8 | 0.795 | 0.794 | 0.791 | 0.79 | 0.785 | 0.782 | 0.798 |
| **Transformer** | **0.818** | **0.817** | **0.815** | **0.813** | 0.81 | **0.808** | **0.802** | **0.802** | **0.798** | **0.797** | 0.793 | **0.791** | **0.805** |
| Graph Transformer | 0.808 | 0.808 | 0.804 | 0.803 | 0.8 | 0.799 | 0.793 | 0.79 | 0.789 | 0.787 | 0.784 | 0.782 | 0.796 |
| $\mathbf{MBT_{early}}$ | **0.826** | **0.824** | **0.82** | **0.817** | **0.815** | **0.811** | **0.806** | **0.805** | **0.802** | **0.799** | 0.797 | **0.794** | **0.81** |
| $MBT_{mid}$ | 0.821 | 0.819 | 0.816 | 0.815 | 0.812 | 0.809 | 0.805 | 0.804 | 0.799 | 0.798 | 0.794 | 0.792 | 0.807 |
| $MT_{early}$ | 0.814 | 0.817 | 0.815 | 0.814 | 0.809 | 0.808 | 0.804 | 0.802 | 0.798 | 0.798 | 0.795 | 0.793 | 0.806 |
| $MT_{mid}$ | 0.819 | 0.82 | 0.818 | 0.816 | 0.813 | 0.81 | 0.805 | 0.804 | 0.799 | 0.796 | 0.793 | 0.791 | 0.807 |
| $BCMAT_{early}$ | 0.811 | 0.813 | 0.815 | 0.806 | 0.814 | 0.81 | 0.796 | 0.808 | 0.788 | 0.796 | **0.802** | 0.791 | 0.804 |
| $BCMAT_{mid}$ | 0.819 | 0.817 | 0.814 | 0.811 | 0.809 | 0.807 | 0.802 | 0.801 | 0.797 | 0.796 | 0.793 | 0.791 | 0.805 |
| $MBT_{early}NCE_{EFN}$ | 0.807 | 0.806 | 0.808 | 0.804 | 0.807 | 0.803 | 0.803 | 0.798 | 0.793 | 0.797 | 0.792 | 0.797 | 0.8 |
| $MBT_{early}Cosine_{EFN}$ | 0.812 | 0.808 | 0.811 | 0.808 | 0.811 | 0.809 | 0.807 | 0.801 | 0.798 | 0.802 | 0.796 | 0.793 | 0.805 |
| $\mathbf{MBT_{early}L2_{EFN}}$ | **0.871** | **0.845** | **0.851** | **0.856** | **0.841** | **0.857** | **0.844** | **0.867** | **0.852** | **0.836** | **0.842** | **0.843** | **0.851** |
| $MBT_{early}L2_{RFN}$ | 0.834 | 0.827 | 0.834 | 0.828 | 0.827 | 0.833 | 0.83 | 0.838 | 0.838 | 0.823 | 0.819 | 0.827 | 0.829 |
| $MBT_{early}L2_{FC}$ | 0.819 | 0.815 | 0.816 | 0.814 | 0.81 | 0.808 | 0.802 | 0.802 | 0.797 | 0.796 | 0.793 | 0.791 | 0.805 |

For fusing EHR text data to EHR numeric, early fusion outperforms other strategies. Particularly, feature concatenation (MT) benefits mortality prediction the most, whereas MBT improves the vasopressor need task the most.

## 3.2 Self-supervised predictive regularization using $L_2$ loss with normalized context vector is crucial

As shown in Table 2, SSL regularization using $\mathcal{L}_2$ loss with normalized context vector $c_t$ by LN performs the best for both prediction tasks. Note that the performance gap between $MT_{early}L2_{EFN}$ and $MT_{early*}L2_{EFN}$ indicates the importance of context vector normalization. Note that the L2 loss yields comparatively larger values than the other auxiliary losses, i.e., cosine and NCE. As a result, incorporating an additional normalization process balance the auxiliary L2 loss with supervised loss, thus improving the model performance. Moreover, SSL with encoded future data (Fig. 2-(b)), which is introduced in the original CPC paper does not always outperform other alternatives, which we partly connect with the low dimensionality of raw future numeric in Sec. 2.5.

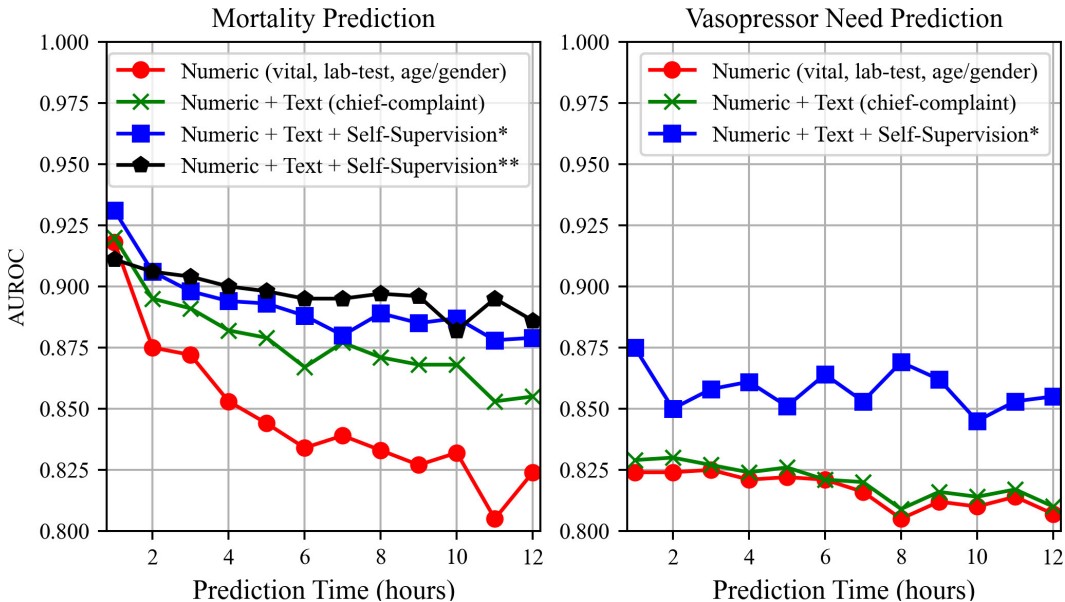

Figure 3: Average test AUROC of mortality (left) and vasopressor need (right) prediction using the best-performing model from each study. * and ** indicates the best performing model from Sec. 2.4 and Sec. 2.5, respectively. Average AUROC are 0.8463, 0.877, 0.8925, 0.8971 for mortality task and 0.8166, 0.8203, 0.858 for vasopressor task (from top to bottom in the legend box).

### 3.3 BOTH BI-MODAL FUSION AND SELF-SUPERVISED PREDICTIVE REGULARIZATION IMPROVES MORTALITY AND VASOPRESSOR NEED PREDICTION

After selecting the best option using validation AUROC for 1) a unimodal network, 2) bimodal fusion strategy, and 3) SSL method respectively, we used the test AUROC to compare their performances. As shown in Fig. 3, both EHR text supplementation and SSL regularization improve the predictive performance of both tasks, i.e., mortality prediction and vasopressor need prediction. Specifically, adding EHR text data to EHR numeric by bi-modal fusion improves overall/far-future (11-12h) prediction of the uni-modal model (baseline) by 0.031/0.031 in mortality prediction and by 0.004/0.003 in vasopressor need prediction. Additional SSL loss further improves overall/far-future prediction of the bi-modal model by 0.020/0.031 in mortality prediction and by 0.038/0.045 in vasopressor need prediction. Note that for the baseline unimodal method, mortality predictive performance degrades much as prediction time gets further in the future, unlike vasopressor need prediction. Though both bimodal fusion and SSL regularization improve the overall predictive performance for mortality prediction, they yield more improvement as prediction time gets further in the future. However, for vasopressor need prediction, the performance gap between near-future prediction and far-future prediction is small, and self-supervision helps overall prediction accuracy much more than text data supplementation.

## 4 CONCLUSION

This paper proposes a novel hourly deterioration prediction model for urgent patients in the ED/ICU. With extensive experiments, we show that both multi-modal fusion and self-supervised predictive regularization effectively improve the performance of mortality and vasopressor need prediction in a fine-grained time resolution; in mortality prediction, both multi-modal fusion and SSL regularization specifically improve the far-future prediction. For vasopressor need prediction, SSL improves not only the far-future prediction but also the overall prediction. In addition, we show the importance of context vector normalization for $L_2$ loss in SSL predictive coding regularization. We believe our method will advance timely intervention and effective resource allocation in the ED/ICU with the improved and thus more trustworthy prediction of patient's critical events.

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

# A APPENDIX

## A.1 SELECTED FEATURES

Our selected six vital-sign data includes heart rate, respiration rate, blood pressure (diastolic and systolic), temperature, and pulse oximetry. The selected 10 lab-tests include Hematocrit, Platelet, WBC, Bilirubin, pH, HCO3, Creatinine, Lactate, Potassium, and Sodium Sung et al. (2021).

## A.2 FEATURE IMPORTANCE

For more detailed information about each numeric feature's influence on the prediction decision, we further calculated the contribution of vital-signs and lab-test on mortality and vasopressor use prediction.

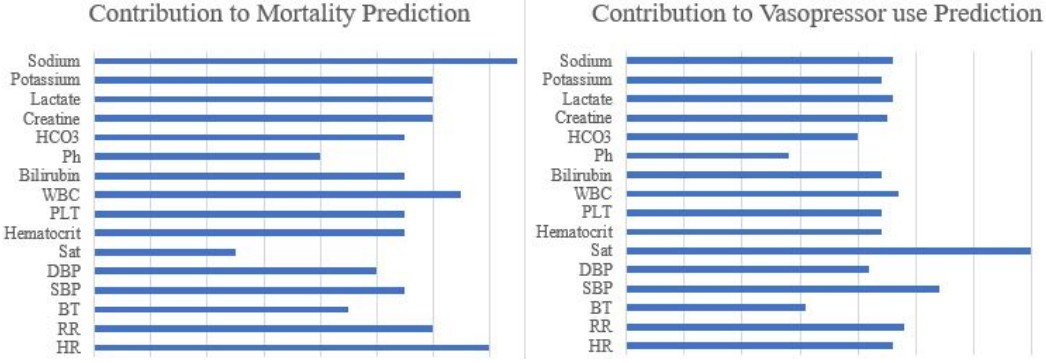

Figure 4: Contribution of vital signal and lab-test features for mortality and vasopressor use prediction with Uni-modal Transformer. We used the Integrated Gradients and averaged 12 future predictions.

## A.3 MODEL COMPLEXITY

Table 3: Number of parameters on each model. $*$ indicates any of the three different loss types of $NCE, Cosine$, or $L_2$

| Models | Number of Parameters (M) |
|---|---|
| GRU-D | 1.03 |
| LSTM | 0.41 |
| Transformer | 6.06 |
| Graph Transformer | 7.16 |
| $MBT_{early}$ | 19.74 |
| $MBT_{mid}$ | 19.74 |
| $MT_{early}$ | 13.74 |
| $MT_{mid}$ | 16.64 |
| $BCMAT_{early}$ | 17.64 |
| $BCMAT_{mid}$ | 18.68 |
| $MT_{early*EFN}$ | 14.53 |
| $MT_{early*RFN}$ | 14.53 |
| $MT_{early*FC}$ | 13.74 |
| $MBT_{early*EFN}$ | 21.31 |
| $MBT_{early*RFN}$ | 21.31 |
| $MBT_{early*FC}$ | 19.74 |

