# OpenReview forum: "Self-Supervised Predictive Coding with Multimodal Fusion for Patient Deterioration Prediction in Fine-grained Time Resolution"
_ICLR.cc/2023/Workshop/TML4H — ICLR 2023 Workshop TML4H Oral_

### Official Review · Reviewer_hDuV · 2023-02-27
**Several questions about the experiments**

**Rating:** 6
**Confidence:** 3

**Review:**

This paper proposed to predict mortality and vasopressor needs from Electronic Health Records (EHR) by self-supervised predictive coding.
Overall this paper is well-written and easy to follow. I just have several questions and comments:

1. As shown in Table 2, using additional normalization would bring huge performance improvements. It would be better if the authors could elaborate more about this with intuitive explanations.
2. In Table 2, I would suggest also highlighting the best AUROC of each hourly range in bold.
3. I notice that among MT_{early}NCR_{EFN}, MT_{early}Cosin_{EFN} and MT_{early}L_{2EFN}, MT_{early}Cosin_{EFN} would perform better than the other two in modality prediction, while for vasopressor need prediction, it seems less effective than MT_{early}L_{2EFN}. Is there any explanations for that?

---

### Official Review · Reviewer_hRUT · 2023-03-02
**The proposed method in this paper utilizes multimodal fusion and self-supervised prediction regularization for hourly deterioration prediction. The approach is interesting and shows promise for predicting deterioration in urgent patients on an hourly basis. However, there are some concerns that need to be addressed in order to further improve the research**

**Rating:** 6
**Confidence:** 4

**Review:**

In this research paper, an hourly prediction method is proposed that utilizes multimodal fusion and self-supervised prediction regularization for predicting hourly deterioration of urgent patients in the ED/ICU. The study compares the performance of unimodal models, bimodal fusion strategies, and self-supervised constraints on deterioration prediction.
While the research is interesting and valuable for clinical practices, there are two concerns that need to be addressed. Firstly, the paper does not provide clear information on the representation learning used for the numerical data. Specifically, it is unclear how numerical values are transformed into vector representations, or if the original values are directly adopted as input to the model.
Secondly, the authors could improve their explanation of the predicted results for each hour in the multimodal fusion approach. For instance, it would be helpful if they could explain which part of the input data is responsible for changes in predicted results from one hour to the next. This additional information would enhance the paper's clinical applicability.
Overall, the paper presents an interesting approach to predicting hourly deterioration in urgent patients, and the concerns raised above could be addressed to further improve the research.

---

### Official Review · Reviewer_9oiK · 2023-03-03
**The paper is well-written. The aim is clear and meaningful, with sufficient experiments and persuasive results.**

**Rating:** 8
**Confidence:** 4

**Review:**

The paper proposes a multi-modal predictor for clinical critical tasks. It explores the fusion of temporal EHR numerical data and text data and improves the pipeline's performance by progressively adding strategies such as self-supervised learning.

Strength:
The paper is well-written, and I enjoy reading it. The authors present sufficient experiments to support their proposed methods, where alternative strategies are also examined, and the results are mostly persuasive.

Weakness:
1. Figure 2 does not clearly illustrate the encoding methods in Section 2.5. More explicit information on how the encoder works (especially b and c) is needed. Please consider adding more introductions in captions.

Suggestion:
I suggest the authors analyze the feature importance of EHR numerical data. As they are actually interpretable signs, the feature importance analysis can further facilitate clinical applications.

---

### Meta-Review · Area_Chair_NXBX · 2023-03-05

**Recommendation:** Accept (Poster)
**Confidence:** 5

**Metareview:**

The paper proposes a self-supervised predictive coding approach to predict mortality and vasopressor needs from EHR data. The overall quality of the paper is good, but the reviewer has some questions and comments for improvement. For example, the authors could elaborate more on the additional normalization used in Table 2 to achieve significant improvements in performance.

In the final version, the authors are highly encouraged to address these concerns to improve the quality of this paper.